DISCOVERY REPORT

# The Gr64 cluster of gustatory receptors promotes survival and proteostasis of epithelial cells in *Drosophila*

**Michael E. Baumgartner**[1☯¤a], **Alex Mastrogiannopoulos**[1☯], **Iwo Kucinski**[2¤b],
**Paul F. Langton**[1], **Eugenia Piddini**[1]*

**1** School of Cellular and Molecular Medicine, University of Bristol, Bristol, United Kingdom, **2** The Wellcome Trust/Cancer Research UK Gurdon Institute and Zoology Department, University of Cambridge, Cambridge, United Kingdom

☯ These authors contributed equally to this work.
¤a Current address: The Perelman School of Medicine at the University of Pennsylvania, Philadelphia, Pennsylvania, United States of America
¤b Current address: Wellcome & MRC Cambridge Stem Cell Institute and Department of Haematology, University of Cambridge, Cambridge, United Kingdom
* eugenia.piddini@bristol.ac.uk

## Abstract

Gustatory Receptor 64 (Gr64) genes are a cluster of 6 neuronally expressed receptors involved in sweet taste sensation in *Drosophila melanogaster*. Gr64s modulate calcium signalling and excitatory responses to several different sugars. Here, we discover an unexpected nonneuronal function of Gr64 receptors and show that they promote proteostasis in epithelial cells affected by proteotoxic stress. Using heterozygous mutations in ribosome proteins (*Rp*), which have recently been shown to induce proteotoxic stress and protein aggregates in cells, we show that *Rp/+* cells in *Drosophila* imaginal discs up-regulate expression of the entire Gr64 cluster and depend on these receptors for survival. We further show that loss of Gr64 in *Rp/+* cells exacerbates stress pathway activation and proteotoxic stress by negatively affecting autophagy and proteasome function. This work identifies a noncanonical role in proteostasis maintenance for a family of gustatory receptors known for their function in neuronal sensation.

## Introduction

Gustatory Receptors 64 (Gr64s) are a group of 6 tandem gustatory receptor genes (*a* through *f*) involved in mediating sensation of sugars, fatty acids, and glycerol in the adult nervous system [1–4]. Gr64s are thought to sense distinct ligands via distinct mechanisms: Gr64e, for example, is reported to act as a ligand gated ion channel in response to glycerol binding, whereas it acts downstream of phospholipase C in fatty acid sensation [4]. Gr64s are thought to function as heterodimers with each other and alternate gustatory receptors, as, for example, both Gr64a and Gr64f are required to mediate responses to certain sugars [5].

Mutations in ribosomal proteins or ribosome biogenesis factors can result in a class of disorders known as ribosomopathies [6–8]. While roles have been established for nucleolar stress,

**Data Availability Statement:** All data are provided within this manuscript and its supplementary files. Numerical data, along with the statistical tests run, are provided in the supplementary file entitled S1_Data.xlsx.

**Funding:** This work was supported by a Cancer Research UK Programme Foundation Award to EP (Grant C38607/A26831) and a Wellcome Trust Senior Research Fellowship to EP (205010/Z/16/Z). The funders had no role in study design, data collection and analysis, decision to publish, or preparation of the manuscript.

**Competing interests:** The authors have declared that no competing interests exist.

**Abbreviations:** Gr64, Gustatory Receptor 64; ISR, integrated stress response; RNAi, RNA interference; Rp, ribosome protein.

p53 activation, and translational defects in disease progression, ribosomopathy etiology remains poorly understood [8]. Ribosome protein (*Rp*) mutations are well studied in *Drosophila*, wherein the majority of cytosolic *Rp* encoding genes yield the so-called "Minute" phenotype when heterozygous mutant [9]. *Rp/+* flies are viable and fertile but exhibit a developmental delay [10]. Epithelial cells in *Rp/+* larvae exhibit reduced translation rates, increased cell-autonomous apoptosis, and stress pathway activation, including, JNK, JAK/STAT, Toll/IMD signalling, and the oxidative stress response [11–15]. *Rp/+* cells also undergo cell competition and are eliminated from the tissue when confronted with wild-type cells in mosaic tissues [16,17]. *Rp/+* cells are therefore said to behave as "losers" relative to wild-type "winners."

*Rp/+* cells have recently been shown to suffer from chronic proteotoxic stress. They exhibit proteasome and autophagy defects, activation of the integrated stress response (ISR), a stoichiometric imbalance of large and small ribosomal subunit proteins and an accumulation of intracellular protein aggregates [18–21]. Importantly, boosting proteostasis rescues stress pathway activation, cell autonomous apoptosis, and competitive elimination [19,20]. These findings point to proteotoxic stress as a potential driver of the pathologies associated with ribosomopathies.

By characterising the biology of *Rp/+* cells, we identify a new function for *Gr64* genes in maintaining proteostasis in epithelial cells. We find that *Gr64* genes, whose expression is normally observed in neuronal cells, become up-regulated in wing disc epithelial cells upon *Rp/+* mutation. Loss of Gr64 drives substantial apoptosis in noncompeting and competing *Rp/+* cells and exacerbates stress pathway activation. Furthermore, loss of Gr64 exacerbates proteotoxic stress in *Rp/+* cells, by reducing proteasome and autophagy function. Calcium imaging reveals reduced calcium activity in *Rp/+* cells upon Gr64 down-regulation (as measured by the frequency of calcium flashes, [22]), suggesting that Gr64's proteostasis promoting effects might be mediated by calcium signalling.

## Results and discussion

Mining the list of genes differentially expressed in cells heterozygous mutant for the ribosomal protein *RpS3*, we observed up-regulation of all 6 Gr64 gustatory receptors relative to wild-type cells (Fig 1A and [11]). Gr64s were also up-regulated in cells mutant in *mahjong* (*mahj*), an E3 ubiquitin ligase, whose mutation also leads to proteotoxic stress and cell competition [11,19,23,24]. This was conspicuous, as *Gr64s* have no known nonneuronal nor larval function. In order to explore the role of *Gr64s* in cells affected by proteotoxic stress, we tested the effect of removing one copy of the *Gr64* locus on $RpS3^{+/-}$ larvae, using a deficiency spanning the Gr64 locus, along with rescuing constructs of other affected genes (*ΔGr64*) [1]. *ΔGr64/ΔGr64* flies present with no known phenotypes other than deficient gustatory responses [1]. $RpS3^{+/-}$, $ΔGr64^{+/-}$ wing discs, however, exhibited a marked increase in apoptosis over levels seen in wing discs carrying either mutation alone (S1A–S1D Fig), indicating that $RpS3^{+/-}$ cells are acutely reliant on *Gr64s* for their survival. We confirmed this result using a precise CRISPR/Cas-9 deletion of the Gr64 cluster, *Gr64af* [4] (Fig 1B and 1C). *Gr64* did not contribute to survival in the non-Minute context, as non-Minute wing discs homozygous null for *Gr64* exhibited no increase in apoptosis relative to the wild type (S1E and S1F Fig). This is consistent with the fact that *Gr64s* appear to be only minimally expressed in wing discs (Flybase and [11]). Importantly, Gr64 mutations also led to increased apoptosis in wing discs heterozygous mutant for *RpS17* or *RpS23*, 2 other *Rp* genes (S1G–S1J Fig), indicating a general requirement for *Gr64* for multiple *Rp* mutants. *RpS23* discs homozygous mutant for *Gr64* further showed morphological defects with a loss of the characteristic pouch and hinge folds observed

**A**

| Gene name | RpS3[Plac92] $^{+/-}$ vs. WT | | RpS3* $^{+/-}$ vs. WT | | mahj $^{-/-}$ vs. WT | |
|---|---|---|---|---|---|---|
| | FC | p | FC | p | FC | p |
| Gr64a | 6.793 | 1.72E-02 | 10.079 | 3.76E-06 | 9.747 | 4.48E-03 |
| Gr64b | 8.126 | 1.65E-03 | 11.393 | 1.97E-07 | 21.711 | 4.02E-06 |
| Gr64c | 10.218 | 2.25E-05 | 11.814 | 2.36E-09 | 21.687 | 1.16E-05 |
| Gr64d | 10.159 | 1.58E-04 | 13.287 | 1.75E-12 | 11.519 | 5.30E-03 |
| Gr64e | 11.375 | 1.13E-05 | 16.343 | 1.15E-15 | 12.242 | 1.59E-03 |
| Gr64f | 16.366 | 5.84E-10 | 27.968 | 2.38E-21 | 18.160 | 3.43E-04 |

**Fig 1. Noncompeting RpS3$^{+/-}$ cells depend on Gr64 for their survival.** (A) Fold change differences in Gr64 transcript expression relative to WT in wing discs heterozygous mutant for RpS3 (as detected in 2 separate mutant alleles: RpS3{Plac92} or RpS3*) or homozygous mutant for mahjong (mahj). Numbers and p-

values are derived from [11]. (**B**) Wing discs heterozygous for a precise deletion of the *Gr64* cluster (*Gr64af*) (left panel), heterozygous mutant for *RpS3* (middle panel), or heterozygous mutant for both (right panel), assessed for cell death via immunostaining for cleaved-Dcp1 (red). (**C**) Quantification of cell death from wing discs of the same genotypes as in (**B**) ($n_{Gr64}$ = 9, $n_{RpS3}$ = 10, $n_{RpS3,Gr64}$ = 9, 2-sided Mann–Whitney U test). (**D**) Representative image of a wing disc heterozygous mutant for *RpS3* and *Gr64* expressing the *UAS-Gr64abcd-GFP-f* construct driven by *hhGal4* and stained with anti-Ci (cyan) to label the anterior compartment, and with anti-cleaved-Dcp1 (red). (**E**) Wing discs heterozygous mutant for *RpS3* (left panel), heterozygous for both *RpS3* and *Gr64* (middle panel), or heterozygous for both and expressing the *UAS-Gr64abcd-GFP-f* construct, without Gal4 driver (right panel), assessed for cell death by immunostaining for cleaved-Dcp1 (red). (**F**) Quantification of cell death from wing discs of the same genotypes as in (**E**) ($n_{RpS3}$ = 10, $n_{RpS3,Gr64}$ = 10, $n_{RpS3,Gr64,}$ $_{UAS-Gr64}$ = 15, 2-sided Mann–Whitney U test). (**G**) WT (left panel) or *RpS3$^{+/-}$* wing discs (right panel) expressing *Gr64f-RNAi* in the posterior compartment driven by *hh-Gal4* and assessed for cell death with a staining for cleaved caspase3 (red). (**H**) Quantification of cell death from wing discs of the same genotype as in (**G, right panel**) ($n$ = 12, 2-sided Wilcoxon signed rank test). Horizontal lines in C and F indicates the median. For this and all other figures, scale bars correspond to 50 μm, and white dashed lines denote compartment boundaries, where the anterior compartment is shown on the left side of the image and ventral is up. Numerical data can be found in the "Fig 1" sheet of S1 Data. Gr64, Gustatory Receptor 64; RNAi, RNA interference; Rp, ribosome protein; WT, wild-type.

in wing discs (S1I and S1J Fig). To confirm that the effect observed was specifically caused by the *Gr64* mutations, we sought to rescue the effect by overexpressing Gr64 in *RpS3$^{+/-}$*, *Gr64$^{+/-}$* mutants using a UAS construct driving expression of 5 of the 6 Gr64 genes (Gr64abcd-GFP-f) [1]. Overexpression of Gr64 in the P compartment resulted in toxicity and wiped out the compartment (Fig 1D), probably due to unphysiologically high expression levels. However, this specific *UAS-Gr64* transgene has been shown to provide rescuing activity in the absence of Gal4 drivers [1]. Accordingly, we tested whether we could similarly observe rescue of *Gr64$^{+/-}$* induced death by using the rescuing *UAS-Gr64* transgene in the absence of Gal4. Indeed, *UAS-Gr64* substantially rescued *Gr64$^{+/-}$* induced death in *RpS3$^{+/-}$* cells, confirming that the observed effect is due to the *Gr64* mutation (Fig 1E and 1F).

To determine whether the survival function of Gr64 reflects a systemic or cell autonomous role, we knocked down Gr64 specifically in the posterior compartment, using the *hedgehog-Gal4* driver. We used a RNA interference (RNAi) line against Gr64f, which, given that the *Gr64* cluster locus is polycistronic [1], is likely to silence multiple Gr64s. *Gr64f-RNAi* expression in wild-type discs yielded no appreciable change in levels of apoptosis (Fig 1G), whereas expression of *Gr64f-RNAi* in *RpS3$^{+/-}$* wing discs caused a strong increase in apoptosis, specifically in the RNAi expressing cells (Fig 1G and 1H). A modest level of cell death was also observed in the non-RNAi compartment, perhaps due to apoptosis-induced apoptosis or to a systemic effect of *Gr64* silencing. Expression of *Gr64f-RNAi* in clones in *RpS3$^{+/-}$* discs yielded a similar result (S2A–S2C Fig). These data argue that *RpS3$^{+/-}$* cells are cell autonomously dependent on Gr64 for their survival, although an additional non-cell autonomous contribution cannot be ruled out.

Next, we asked whether the requirement of Gr64 in *Rp/+* cells is also observed in other imaginal discs. Both haltere (S2D–S2F Fig) and leg discs (S2G–S2I Fig) showed increased death when Gr64 was knocked down. This effect, however, was less pronounced than in wing discs. Eye discs instead did not appear to be affected by single copy removal of Gr64 (S2J–S2L Fig). Thus, Gr64 is required for *RpS3$^{+/-}$* survival across many but not all imaginal discs.

Having established a prosurvival role for *Gr64* in noncompeting *RpS3$^{+/-}$* cells, we then tested whether *Gr64*s contribute to the survival of competing *RpS3$^{+/-}$* losers. We therefore generated *RpS3$^{+/-}$* losers competing against wild-type winners in wing discs carrying heterozygous mutations in one of any of the 6 *Gr64* genes [25] (Fig 2). Strikingly, heterozygosity for any *Gr64* yielded a substantial increase in *RpS3$^{+/-}$* loser cell death at the winner/loser interface (Fig 2A–2H). Furthermore, loser cell clones were smaller in wing discs carrying mutations in *Gr64b*, *Gr64c*, *Gr64d*, *Gr64e*, or *Gr64f*, but not *Gr64a* (Fig 2I). *RpS3$^{+/-}$* loser cells are therefore dependent on *Gr64*s in both noncompetitive and competitive conditions. While these results might suggest that each specific Gr64 protein contributes to *RpS3$^{+/-}$* survival and these constructs have been used successfully in screens of Gr64 gustatory function [2], because of the

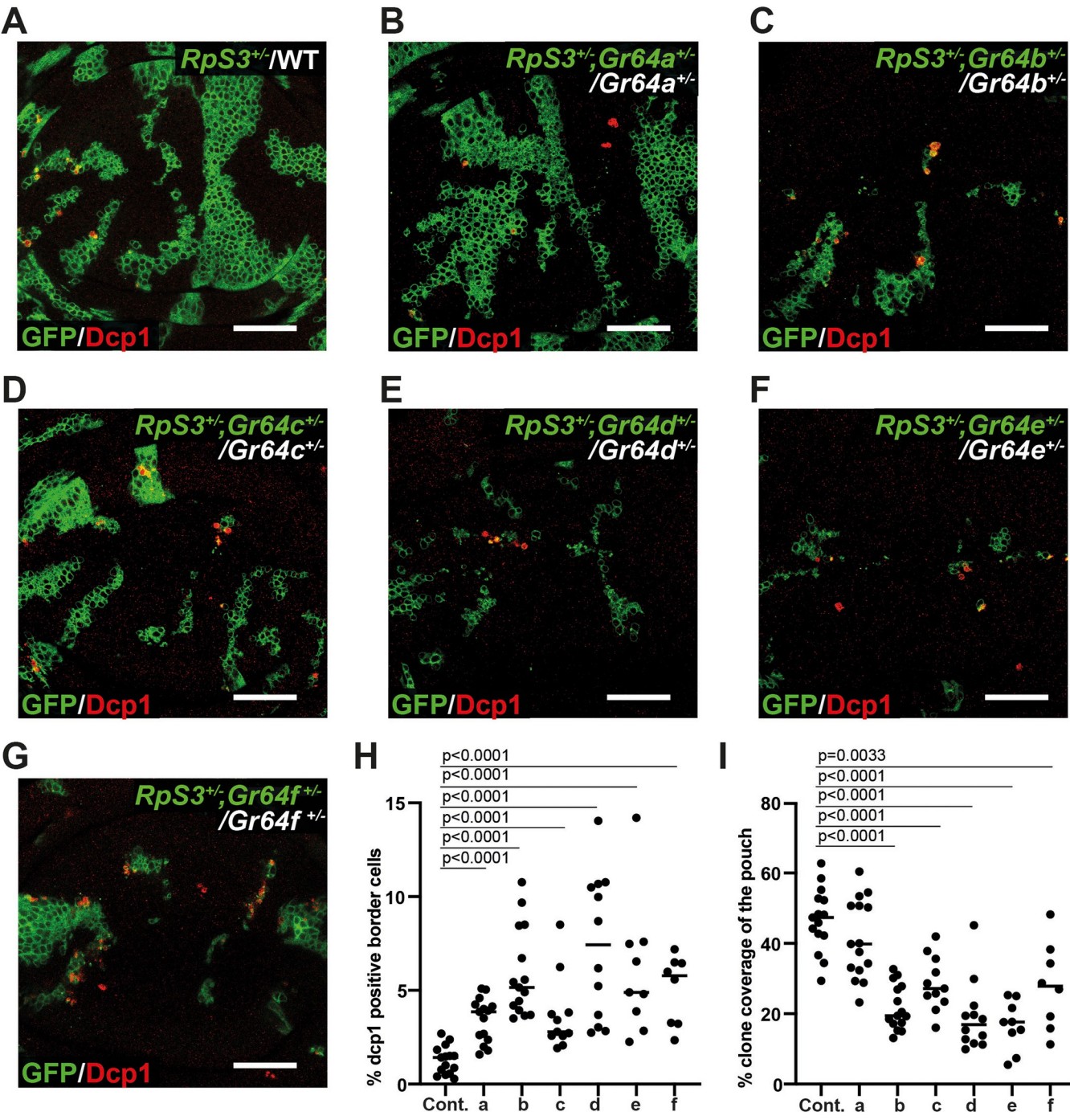

**Fig 2. Heterozygosity at Gr64 loci exacerbates competitive *RpS3*⁺ᐟ⁻ loser cell elimination.** (A–G) Representative images of wing discs containing *RpS3*⁺ᐟ⁻ loser cells (green) competing against wild-type winners (unlabelled) and stained for cleaved-Dcp1 (red). *RpS3*⁺ᐟ⁻ clones were generated in a wild-type background (**A**) or in wing discs heterozygous for any one of the Gr64 genes a through f (mutations used were $Gr64a^{GAL4}$, $Gr64b^{LEXA}$, $Gr64c^{LEXA}$, $Gr64d^1$, $Gr64e^{LEXA}$, and $Gr64f^{LEXA}$) (**B–G**). (**H**) Quantification of the percentage of cells undergoing apoptosis at loser clone borders in wing discs of genotypes as shown in (**A–G**). Statistics reflect multiple logistic regression across 3 replicates (details provided in Materials and methods). (**I**) Quantification of loser cell growth in wing discs of the genotypes shown in (**A–G**), as measured by the percent loser clone coverage of the pouch. Statistics reflect Student $t$ test with FDR p-correction. $n_{control} = 16$, $n_{Gr64aGAL4} = 15$, $n_{Gr64bLEXA} = 15$, $n_{Gr64cLEXA} = 11$, $n_{Gr64d[1]} = 12$, $n_{Gr64eLEXA} = 9$, $n_{Gr64fLEXA} = 8$. For all quantifications, the horizontal line indicates the mean. Scale bars correspond to 50 μm. Numerical data can be found in the "Fig 2" sheet of S1 Data. FDR, false discovery rate; Gr64, Gustatory Receptor 64; Rp, ribosome protein.

polycistronic nature of the locus, it is difficult to draw conclusions on the requirements of individual Gr64 isoforms here, as mutations in one coding sequence might affect the regulation and behaviour of the others.

*Rp/+* mutations cause stress pathway activation and cellular malfunctions, many of which are linked with the loser status [11,12,18–21,26]. To evaluate how *Gr64* loss affects cellular stress responses, we identified milder expression conditions for *Gr64f-RNAi*, using the posterior compartment-specific *engrailed-Gal4* driver. With these conditions, Gr64f knockdown in noncompeting *RpS3*$^{+/-}$ cells exhibited a comparatively mild increase in apoptosis (Fig 3A and 3B), allowing us to investigate *Gr64* function without widespread cell death. *Gr64f-RNAi* yielded increased activation of the oxidative stress response, as measured by *GstD1-GFP* reporter expression [27] (Fig 3C and 3D). *Gr64f-RNAi* did not yield an increase in *GstD1-GFP* signal in wild-type wing discs (S3A and S3B Fig), demonstrating that this result is specific to the *RpS3*$^{+/-}$ context. Consistent with these results, an increase in *GstD1-GFP* expression was also observed in *RpS3*$^{+/-}$ wing discs heterozygous for *ΔGr64* (S3C and S3D Fig). Furthermore, *Gr64f-RNAi* expression in *RpS3*$^{+/-}$ discs resulted in a mild increase in JNK pathway and in the ISR, as measured by immunostaining for phosphorylated JNK (Fig 3E and 3F) and phosphorylated eIF2α (Fig 3G and 3H), respectively. No differences in phospho-JNK (S4A and S4B Fig) or phospho-eIF2α (S4C–S4D Fig) levels were seen in the posterior compartments of otherwise genetically identical wing discs lacking the *Gr64f-RNAi*, confirming that this result is due to RNAi expression. These data indicate that loss of Gr64 exacerbates stress responses seen in *Rp/+* cells.

We next investigated how *Gr64* alleviates stress signalling and improves viability of *Rp/+* cells. We previously reported that the ISR and oxidative stress response in *RpS3*$^{+/-}$ cells are driven by accumulation of protein aggregates, accompanied by reduced degradation of proteins by the proteasome and by autophagy, leading to sustained proteotoxic stress [19]. Thus, we tested the effect of Gr64 reduction on proteostasis pathways using ProteoFlux and ReFlux, genetically encoded reporters of proteasome and autophagy function, respectively, which we previously developed [19]. In these reporters, GFP is tagged for degradation through the proteasome or the autophagosome by fusion to CL1 or ref(2)P, respectively. These GFP fusions are expressed under the control of a heat shock promoter, which enables heat-shock induced pulse-chase experiments, whereby decline in GFP signal is a quantitative measure of proteasomal or autophagosomal degradation rates [19]. Using these tools, we found that *Gr64f-RNAi* causes a further reduction in proteasomal (Fig 4A–4C) and autophagosomal (Fig 4D–4F) degradation rates, beyond the defects already present in *RpS3*$^{+/-}$. These data indicate that loss of Gr64 yields a substantial impairment in cellular proteolytic pathways. Furthermore, *Gr64f-RNAi* yields an increase in ref(2)P-positive foci (Fig 4G), structures that we have previously shown to colocalise with poly-ubiquitinylated aggregates in *RpS3*$^{+/-}$ cells [19]. We conclude that Gr64 loss causes increased stress signalling and cell death by exacerbating proteotoxic stress.

Worsening of proteotoxic stress in ribosome mutants could derive from an inhibition of protein catabolic processes or from an increase in protein translation, which, by producing more proteins, could cause a further burden on proteostasis. Interestingly, OPP, a global translation reporter, revealed a mild but statistically significant decrease in translation upon expression of *Gr64f-RNAi* (Fig 4H and 4I). This is consistent with the observed increase in eIF2α-phosphorylation (Fig 3G and 3H), which is a translation inhibitor as well as a stress pathway marker [28]. Thus, the increase in proteotoxic stress observed upon Gr64 reduction is not due to increased translation but rather to a failure to clear defective proteins.

How could a family of taste receptors play a role in proteostasis? Gr64 proteins are sweet tastant receptors that cause an increase in cytoplasmic calcium upon ligand binding [25].

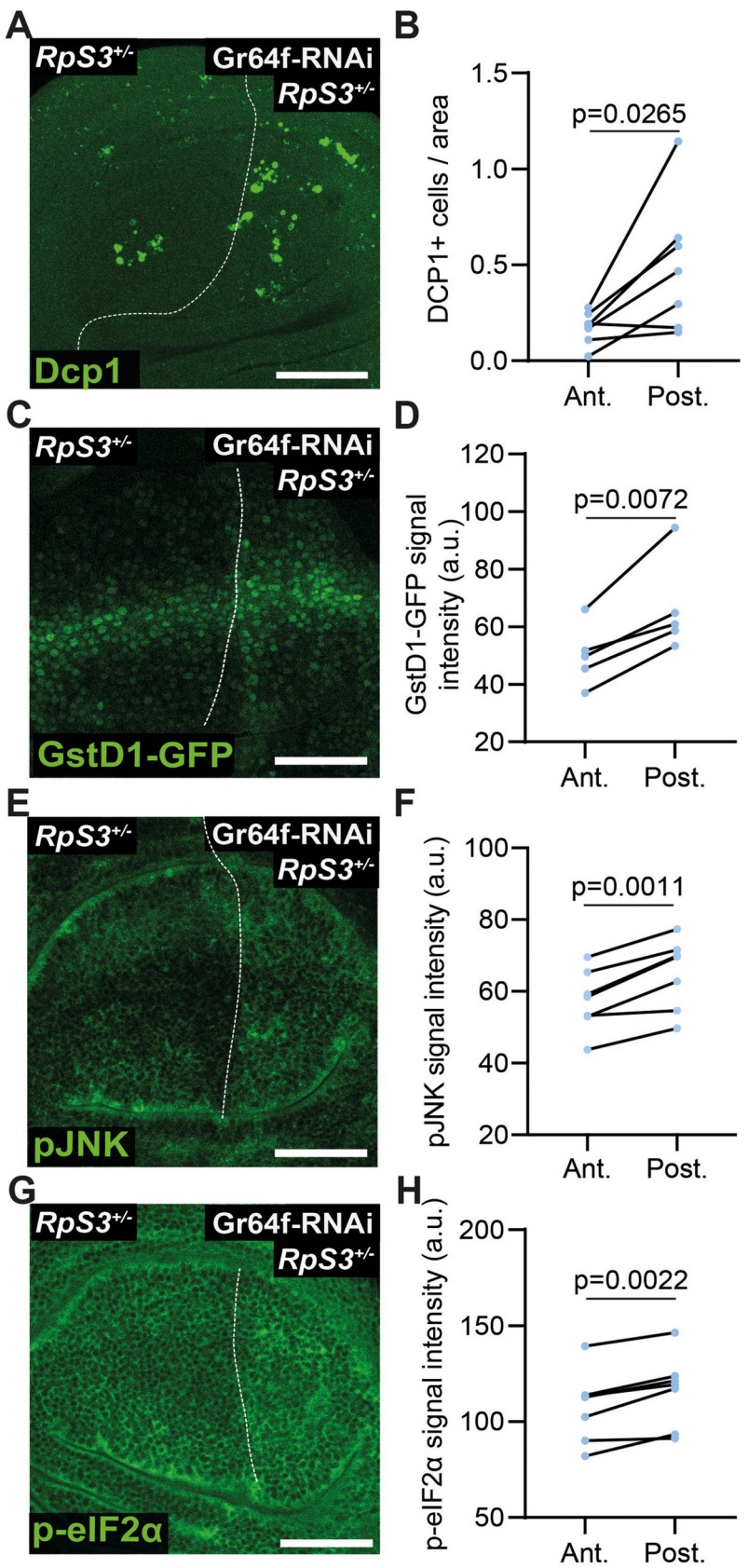

**Fig 3. Loss of Gr64 worsens stress pathway activation in *RpS3*$^{+/-}$.** (**A, B**) *RpS3*$^{+/-}$ wing discs expressing *Gr64f-RNAi* driven by *enGal4* in the posterior compartment stained for cell death, as detected by cleaved Dcp1 (green) (**A**) along with quantification in (**B**) ($n = 7$, 2-sided Wilcoxon signed rank test). (**C, D**) GstD1-GFP reporter expression (green) in *RpS3*$^{+/-}$ wing discs expressing *Gr64f-RNAi* driven by *enGal4* in the posterior compartment (**C**) along with quantification in (**D**) ($n = 6$, 2-sided paired *t* test). (**E, F**) *RpS3*$^{+/-}$ wing discs expressing *Gr64f-RNAi* driven by *enGal4* in the posterior compartment and stained for phosphorylated JNK (green) (**E**) along with quantification in (**F**) ($n = 7$, 2-sided paired *t* test). (**G, H**) *RpS3*$^{+/-}$ wing discs expressing *Gr64f-RNAi* driven by *enGal4* in the posterior compartment and stained for phosphorylated eIF2α (green) (**G**) along with quantification in (**H**) ($n = 7$, 2-sided paired *t* test). Scale bars correspond to 50 μm. Numerical data can be found in the "Fig 3" sheet of S1 Data. Gr64, Gustatory Receptor 64; RNAi, RNA interference; Rp, ribosome protein.

Calcium is known to play a role in modulating proteostatic pathways, [29,30] suggesting that Gr64 could ameliorate proteostasis by modifying calcium signalling. We therefore investigated what effect loss of Gr64 might have on calcium signalling using a GCaMP fluorescent calcium reporter [31].Consistent with this hypothesis, we found that silencing Gr64 in the posterior compartment of *RpS3*$^{+/-}$ wing discs reduced the frequency of calcium flashes in posterior compartment cells, relative to reference *RpS3*$^{+/-}$ anterior cells, which served as internal control (Fig 4J, S1 Movie).

In this study, we have identified Gr64 taste receptors as novel players in proteostasis control and as cytoprotective regulators in epithelial cells affected by proteotoxic stress. Our data suggest that Gr64s contribute to proteostasis by promoting protein catabolism rather than by inhibiting translation (Fig 4K). This is potentially mediated by calcium signalling, as Gr64 activity typically induces calcium release [25], and calcium is involved in protein folding, the ISR, proteasome, and autophagy function [29,32–34]. Indeed, consistent with this hypothesis, our data indicate that removal of Gr64 modifies Calcium signalling in *Rp/+* cells. It remains to be determined, however, whether this phenotype is causal or a consequence of the proteostasis defects observed in Gr64 knockdown conditions, and thus further work must be done to elucidate Gr64's mechanism of action.

Taste receptors have been implicated in nontaste-related chemosensation in neuronal and neuroendocrine cells both in flies and in mammals [2,35–37]. However, a role in proteostasis and a function in epithelia have not previously been described for taste receptors. Interestingly, dysregulation of olfactory and gustatory receptors has been observed in nonolfactory human brain tissue from individuals suffering from protein-aggregate driven neurodegenerative disorders, including Alzheimer disease, Parkinson disease, and Creutzfeld–Jacob disease as well as in a mouse model of Alzheimer disease [38], and olfactory receptors are expressed near to amyloid plaques in a mouse model of Alzheimer disease [39]. It is therefore possible that gustatory and olfactory receptors play a conserved role in promoting proteostasis in both neuronal and nonneuronal cells.

## Materials and methods

### Fly husbandry and stocks

*Drosophila* lines were kept in an incubator set to 25°C and reared on food prepared according to the following recipe: 7.5 g/L agar powder, 50 g/L baker's yeast, 55 g/L glucose, 35 g/L wheat flour, 2.5% nipagin, 0.4% propionic acid, and 1.0% penicillin/streptomycin. All larvae were dissected at the wandering third instar stage. For experiments with heat-shock–induced clones, vials were transferred to a water bath set to 37°C for 25 minutes on day 3 after egg laying. The vials were then immediately returned to the 25°C incubator and allowed to grow as normal for 3 more days prior to dissection. Only female larvae were dissected.

The following *Drosophila melanogaster* lines were obtained from the Bloomington Drosophila Resource Center: *RpS3{Plac92}* (Cat#BL5627), *RpS3[*]* (Cat#BL5699), *en-Gal4*,

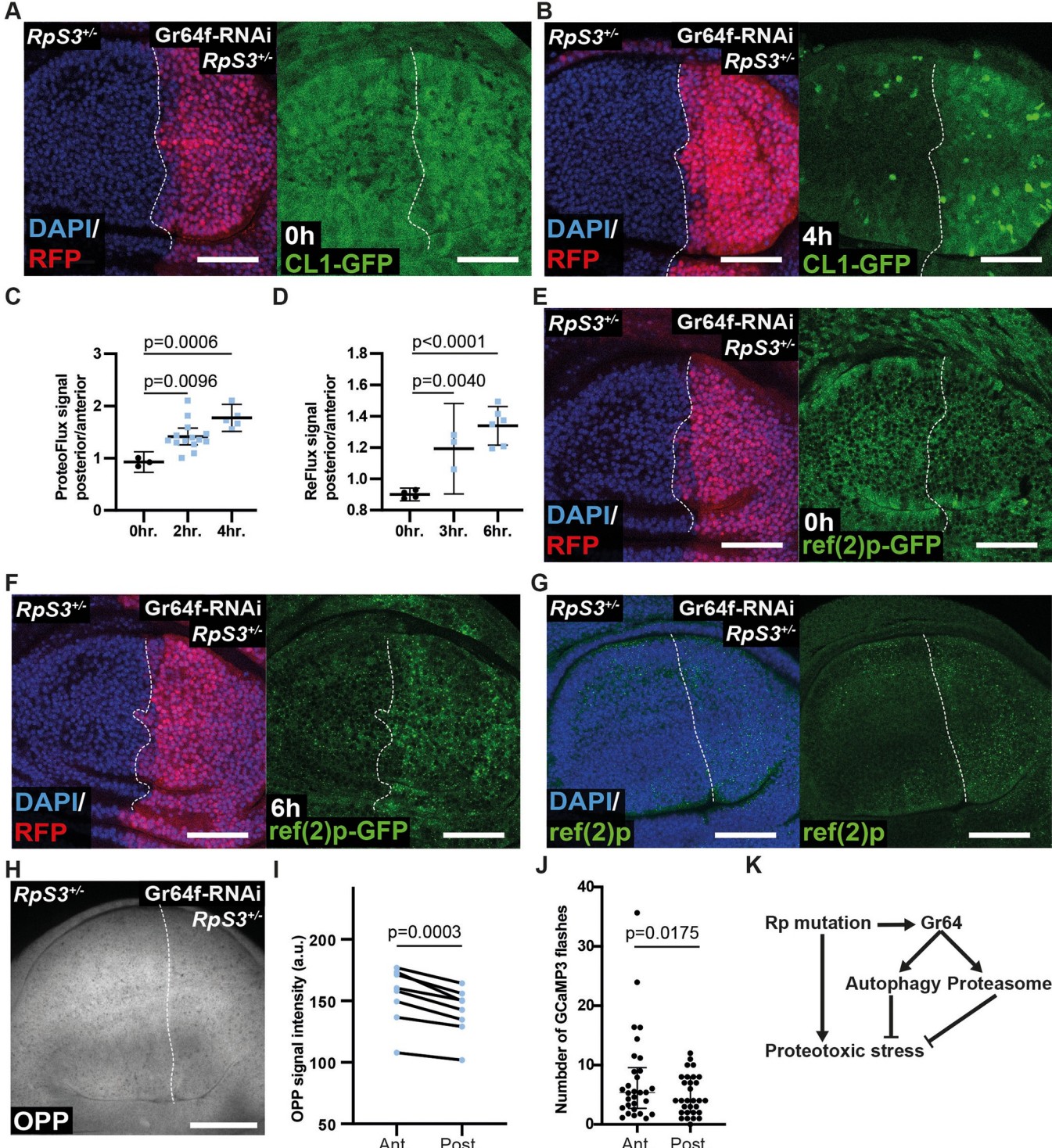

**Fig 4. Loss of Gr64 exacerbates proteostasis defects in $RpS3^{+/-}$.** (**A–C**) $RpS3^{+/-}$ wing discs expressing $Gr64f$-$RNAi$ driven by $enGal4$ in the posterior compartment, marked with RFP (red) and expressing the CL1-GFP/ProteoFlux construct (green), 0 hours (**A**) or 4 hours (**B**) after heat shock along with quantification in (**C**) ($n_{0hr} = 3$, $n_{2hr} = 14$, $n_{4hr} = 5$, 2-sided paired $t$ test, the horizontal line indicates the mean and the whiskers reflect the 95% confidence interval). (**D–F**) $RpS3^{+/-}$ wing discs expressing $Gr64f$-$RNAi$ driven by $enGal4$ in the posterior compartment, marked with RFP (red) and expressing the ref(2)P-GFP/ReFlux construct (green), 0 hours (**E**) or 6 hours (**F**) after heat shock along with quantification in (**D**) ($n_{0hr} = 4$, $n_{3hr} = 3$, $n_{6hr} = 6$, 2-sided paired $t$ test, the horizontal line indicates the mean, and the whiskers reflect the 95% confidence interval). (**G**) $RpS3^{+/-}$ wing discs expressing $Gr64f$-$RNAi$ driven by $enGal4$ in the posterior compartment stained for ref(2)P (green). (**H, I**) $RpS3^{+/-}$ wing discs expressing $Gr64f$-$RNAi$ driven by $enGal4$ in the posterior compartment and assessed for translation through an OPP translation reporter assay (grey) (**H**) along with quantification in (**I**) ($n = 8$, 2-sided paired $t$ test). (**J**) Quantification of

the number of calcium flashes in anterior and posterior compartments of wing discs heterozygous for *RpS3* that also express *Gr64f-RNAi* in the posterior compartment with *enGal4* ($n_{Ant}$ = 30, $n_{Post}$ = 30, 2-sided Wilcoxon signed rank test; the horizontal line indicates the median and the whiskers the interquartile range). (**K**) Proposed model: heterozygous mutation in a ribosomal protein mutant gene triggers proteotoxic stress, making *Rp/+* cells highly dependent upon the autophagosome and proteasome for proteostasis. Gr64 gene cluster is up-regulated in *Rp/+* cells, where it supports cell survival and inhibits proteotoxic stress by promoting autophagy and proteasome functions, possibly via calcium signalling. Scale bars correspond to 50 μm. Numerical data can be found in the "Fig 4" sheet of S1 Data. Gr64, Gustatory Receptor 64; RNAi, RNA interference; Rp, ribosome protein.

*UAS-RFP* (Cat#BL30557), *en-Gal4* (Cat#BL30564), and *RpS17 $^4$, arm-LacZ, FRT80B/TM6* (BL6358). *Gr64f-RNAi* and *40D-UAS* lines (Cat#v100156 and KK60101, respectively) were obtained from the Vienna Drosophila Resource Centre. *hs-CL1-GFP (ProteoFlux)*, *hs-ref(2) P-GFP (ReFlux)* and *hs-FLP, UAS-CD8-GFP;; FRT82B, RpS3{Plac92}, act>RpS3>Gal4/TM6b* were reported in [19]. The following lines were kindly provided by Hubert Amrein: *R1; R2; ΔGr64* (R1 and R2 are constructs rescuing all other genes flanking the *Gr64* locus affected by the deletion) and *UAS-Gr64abcd-GFP-f* [1]. *Gr64a$^{GAL4}$*, *Gr64b$^{LEXA}$*, *Gr64c$^{LEXA}$*, Gr64d$^1$, *Gr64e$^{LEXA}$*, and *Gr64f$^{LEXA}$*. *Gr64-Gal4/lexA* insertions are both expression reporters and validated null mutants [2,25]. The *hs-FLP;; FRT82B* and *y[1], w[1118]* lines were provided by Daniel St. Johnston. The *hh-Gal4/TM6B* and *RpS23$^{R67K}$* were provided by Jean-Paul Vincent [20]. The *GstD1-GFP* line was described in [27]. The *Gr64af* deletion was described in [4]. The *sqh-GCaMP3* stock was described in [31]. The *tub>CD2>Gal4, UAS-CD8GFP; tub-Gal80ts* stock was provided by Bruce Edgar. The *UAS-Gr64abcd-GFP-f, Gr64af* recombinant was generated in this study.

Genotypes for all experimental crosses are provided in Table 1 below.

## Immunofluorescence

Larvae at the wandering third instar stage were washed once and then dissected in PBS before being immediately transferred to a pre-chilled vial of PBS. Samples were then fixed in 4% formaldehyde in PBS at room temperature for 20 minutes. Samples were then washed 3 times in PBS and then permeabilised in 0.25% Triton X-100 in PBS (PBST). The PBST was then aspirated and replaced with blocking buffer (4% fetal calf serum in PBST) and incubated for 30 minutes at room temperature. Primary antibodies were diluted in blocking buffer, and primary incubations took place overnight at 4°C on a rocker. Samples were then washed 3 times in PBST at room temperature for 3 minutes, followed by a 1-hour incubation with secondary antibody diluted 1:500 in blocking buffer along with 0.5μg/mL DAPI. Secondary antibodies used were Alexa-Fluor 488, 555, or 633 (Molecular Probes Eugene, Oregon, USA). Samples were then again washed 3 times for 10 minutes in PBST before being mounted in VECTA-SHIELD (Vector Laboratories Newark, California, USA) on a borosilicate glass slide (number 1.5, VWR International, Radnor, Pennsylvania, USA).

The antibodies used were Rabbit anti-pJNK pTPpY (1:500, Promega, Madison, Wisconsin, USA, Cat#V93B), Rat anti-Ci(1:1,000, DSHB, Iowa City, Iowa, USA, Cat#2A1), Rabbit anti-Ref(2)P (1:2,000, provided by Tor Erik Rusten [40]), Rabbit anti-cleaved Caspase-3 (1:25,000, Abcam, Cambridge, UK, Cat#13847), Rabbit anti-DCP1 (1:2,500, Cell Signalling, Danvers, Massachusetts, USA, Cat#9578S), Rabbit anti-p-eIF2α (1:500, Cell Signalling, Cat#3398T).

## ProteoFlux and ReFlux pulse-chase assays

On day 6 after egg laying, vials containing third instar larvae carrying the ProteoFlux or ReFlux constructs were transferred to a water bath set to 37°C for 40 or 45 minutes, respectively. Larvae were then immediately dissected and transferred to ice cold 4% formaldehyde in PBS to act as a zero time point. The vials were then returned to the 25°C incubator and dissected at 2 and 4 hours after heat shock for ProteoFlux and 3 and 6 hours after heat shock for ReFlux. Larvae were then fixed and mounted as normal.

**Table 1. Genotypes for all experimental crosses.**

| Figure/panel | Genotype |
|---|---|
| | |
| **Main figures** | |
| Fig 1B | *Gr64af/+* (left) <br> *Frt82B, RpS3{Plac92}, tub-dsRed/+* (middle) <br> *Gr64af/ Frt82B, RpS3{Plac92}, tub-dsRed* (right) |
| Fig 1D | *FRT82B, RpS3{Plac92}, hh-Gal4/Gr64af, UAS-Gr64abcd-GFP-f* |
| Fig 1E | *FRT82B, RpS3{Plac92}, ubi-GFP* (left) <br> *FRT82B, RpS3{Plac92}, ubi-GFP/Gr64af* (middle) <br> *FRT82B, RpS3{Plac92}, ubi-GFP/Gr64af, UAS-Gr64abcd-GFP-f* (right) |
| Fig 1G | *UAS-Gr64f-RNAi(KK)/GstD1-GFP; hh-Gal4/+* (left) <br> *UAS-Gr64f-RNAi(KK)/GstD1-GFP; FRT82B, RpS3{Plac92}, hh-Gal4/+* (right) |
| Fig 2A | *hs-FLP, UAS-CD8-GFP/+;; RpS3{Plac92}, act>RpS3>Gal4/+* |
| Fig 2B | *hs-FLP, UAS-CD8-GFP/+;; RpS3{Plac92}, act>RpS3>Gal4/ Gr64a$^{GAL4}$* |
| Fig 2C | *hs-FLP, UAS-CD8-GFP/+;; RpS3{Plac92}, act>RpS3>Gal4/Gr64b$^{LEXA}$* |
| Fig 2D | *hs-FLP, UAS-CD8-GFP/+;; RpS3{Plac92}, act>RpS3>Gal4/ Gr64c$^{LEXA}$* |
| Fig 2E | *hs-FLP, UAS-CD8-GFP/+;; RpS3{Plac92}, act>RpS3>Gal4/Gr64d$^{1}$* |
| Fig 2F | *hs-FLP, UAS-CD8-GFP/+;; RpS3{Plac92}, act>RpS3>Gal4/Gr64e$^{LEXA}$* |
| Fig 2G | *hs-FLP, UAS-CD8-GFP/+;; RpS3{Plac92}, act>RpS3>Gal4/Gr64f$^{LEXA}$* |
| Fig 3A | *hs-FLP; en-Gal4/UAS-Gr64f-RNAi(KK); FRT82B, RpS3{Plac92}, tub-dsRed/Frt82B* |
| Fig 3C | *hs-FLP; en-Gal4, GstD1-GFP/ UAS-Gr64f-RNAi(KK); FRT82B, RpS3{Plac92}, tub-dsRed/ Frt82B* |
| Fig 3E | *hs-FLP; en-Gal4/ UAS-Gr64f-RNAi(KK);FRT82B, RpS3{Plac92}, tub-dsRed/Frt82B* |
| Fig 3G | *hs-FLP; en-Gal4/ UAS-Gr64f-RNAi(KK);FRT82B, RpS3{Plac92}, tub-dsRed/Frt82B* |
| Fig 4A and 4B | *hs-CL1-GFP (ProteoFlux)/(+ or y); enGal4, UAS-RFP/ UAS-Gr64f RNAi(KK); FRT82B, RpS3 {Plac92}, tub-dsRed/+* |
| Fig 4E and 4F | *hs-ref(2)P-GFP (ReFlux)/(+ or y); enGal4, UAS-RFP/ UAS-Gr64f RNAi(KK); FRT82B, RpS3 {Plac92}, tub-dsRed/+* |
| Fig 4G and 4H | *hs-FLP; en-Gal4/ UAS-Gr64f-RNAi(KK);FRT82B, RpS3{Plac92}, tub-dsRed/Frt82B* |
| Fig 4J | *enG4/ UAS-Gr64f-RNAi(KK); FRT82B, RpS3{Plac92}, tub-dsRed/sqhGCaMP3* |
| **Figure supplements** | |
| S1A Fig | *FRT82B, RpS3{Plac92}, ubi-GFP/+* (left) <br> *R1/+; R2/+; FRT82B, RpS3{Plac92}, ubi-GFP/ΔGr64* (right) |
| S1C Fig | *R1/+; R2/+; FRT82B, ubi-GFP/ΔGr64* (left) <br> *R1/+; R2/+; FRT82B, RpS3{Plac92}, ubi-GFP/ΔGr64* (right) |
| S1E Fig | *w$^{1118}$* (left) <br> *Gr64af/Gr64af* (right) |
| S1G Fig | *w;; RpS17$^{4}$, arm-LacZ, FRT80B/+* (left) <br> *w;; RpS17$^{4}$, arm-LacZ, FRT80B/Gr64af* (right) |
| S1I Fig | *RpS23$^{R67K}$/+* (left) <br> *RpS23$^{R67K}$/+; Gr64af/Gr64af* (right) |
| S2A Fig | *hs-FLP; tub>CD2>Gal4, UAS-CD8GFP/ 40D-UAS; tub-Gal80ts/ FRT82B, RpS3{Plac92}, tub-dsRed* |
| S2B Fig | *hs-FLP; tub>CD2>Gal4, UAS-CD8GFP/ UAS-Gr64f-RNAi(KK); tub-Gal80ts/ FRT82B, RpS3 {Plac92}, tub-dsRed* |
| S2D Fig | *40D-UAS/+; FRT82B, RpS3{Plac92}, hh-Gal4/+* |
| S2E Fig | *UAS-Gr64f-RNAi(KK)/+; FRT82B, RpS3{Plac92}, hh-Gal4/+* |
| S2G Fig | *40D-UAS/+; FRT82B, RpS3{Plac92}, hh-Gal4/+* |
| S2H Fig | *UAS-Gr64f-RNAi(KK)/+; FRT82B, RpS3{Plac92}, hh-Gal4/+* |
| S2J Fig | *RpS23$^{R67K}$/+* |
| S2K Fig | *RpS23$^{R67K}$/+; Gr64af/+* |
| S3A Fig | *UAS-Gr64f-RNAi(KK)/GstD1-GFP; hh-Gal4/+* |

*(Continued)*

**Table 1.** (Continued)

| Figure/panel | Genotype |
|---|---|
| | |
| S3C Fig | *GstD1-GFP/+; FRT82B, RpS3{Plac92}, hh-Gal4/+* |
| S3D Fig | *R1/+; R2/GstD1-GFP; FRT82B, RpS3{Plac92}, hh-Gal4/ΔGr64* |
| S4A Fig | *hs-FLP; en-Gal4/+; FRT82B, RpS3{Plac92}, tub-dsRed/Frt82B* |
| S4C Fig | *hs-FLP; en-Gal4/+; FRT82B, RpS3{Plac92}, tub-dsRed/Frt82B* |
| S1 Movie | *enG4/ UAS-Gr64f-RNAi(KK); FRT82B, RpS3{Plac92}, tub-dsRed/sqhGCaMP3* |

## OPP translation assay

Larvae were washed once and then dissected in pre-warmed Schneider's medium. Hemi-larvae were then transferred to a 1.5 mL Eppendorf tube containing 5 μM OPP (Molecular Probes, Cat#C10456) diluted in Schneider's medium and placed in a heating block set to 25˚C for 15 minutes. Samples were then washed quickly in PBS and then fixed in 4% formaldehyde in PBS for 20 minutes at room temperature, permeabilised for 30 minutes at room temperature in 0.5% PBST and incubated for 30 minutes in blocking buffer. Samples were then washed in PBS, and staining was performed using the Click-iT Plus protocol according to manufacturer's instructions.

## Calcium live imaging

Larvae were dissected in Schneider's medium, and wing discs were placed on a custom imaging chamber filled with Schneider's medium. The chamber was made by attaching a glass coverslip to a steel slide with silicone sealant. The other side of the chamber was covered with an oxygen permeable membrane (YSI Membrane Kit Standard, Yellow Springs, Ohio, USA). Wing discs were then live imaged on a Yokogawa CV7000S microscope using a 20× long working distance 0.45 numerical aperture dry objective. The images taken were XYZT stacks consisting of 3 slices over 6-μm depth, and each stack was imaged every 10 seconds for 10 minutes. The images were manually analysed in Fiji. Flashes in the anterior and posterior compartments were counted and the frequency of flashes normalised to posterior and anterior compartment surface area.

## Imaging, quantification, and statistical analysis

All images were acquired as z-stacks with 1 μM z-planes on Leica SP5 and SP8 confocal microscopes using a 40× 1.3 numerical aperture PL Apo Oil objective. All images were quantified using the PECAn image and data analysis pipeline [41]. Statistical analysis was performed using Rstudio and Graphpad Prism 8 software. Specific statistical tests and number of replicates performed for each experiment is provided in the statistical source data sheet (S1 Data). The following workflow was performed: If data met parametric assumptions (normality, homogeneity of variance), a *t* test or paired *t* test was used. If these criteria were not met, a Mann–Whitney U test or Wilcoxon signed rank test was used. A minimum of 2 independent biological replicates were performed of each experiment. Logistic regression was performed in PECAn. The dependent variable was the number of viable and nonviable $RpS3^{+/-}$ cells in the loser clone border, as determined via a staining for cleaved DCP1. Predictor variables were determined by running the model with different, noncollinear variables (as determined by variance inflation factor below 5) and scoring by minimising the Akaike information criterion. A false discovery rate (FDR) correction was used to correct for multiple comparisons.

## Supporting information

**S1 Fig. (Related to Fig 1). Further characterisation of the Gr64 and RpS3 interaction using additional *Minute* and *Gr64* mutants.** (**A**) Wing discs heterozygous mutant for *RpS3* without (left panel) or with (right panel) a heterozygous deficiency in the *Gr64* locus (*ΔGr64*) and assessed for cell death with a staining for cleaved-caspase3 (red), along with quantification in (**B**) ($n_{RpS3}$ = 6, $n_{RpS3,ΔGr64}$ = 7, 2-sided Mann–Whitney U test). (**C**) Wing discs heterozygous for a deficiency spanning the *Gr64* locus (*ΔGr64*) in a wild type (left panel) or *RpS3*$^{+/-}$ background (right panel) and assessed for cell death with a staining for cleaved-caspase3 (red), along with quantification in (**D**) ($n_{ΔGr64}$ = 10, $n_{RpS3,ΔGr64}$ = 11, 2-sided Mann–Whitney U test). (**E**) Wild type (left panel) or *Gr64* homozygous null (right panel) wing discs assessed for cell death via a staining for cleaved-Dcp1 (red), along with quantification in (**F**) ($n_{wt}$ = 8, $n_{Gr64}$ = 10, 2-sided Mann–Whitney U test). (**G**) Wing discs heterozygous for *RpS17* (left panel) or heterozygous for both *RpS17* and *Gr64* (right panel) assessed for cell death with a staining for cleaved-Dcp1 (red), along with quantification in (**H**) ($n_{Rps17}$ = 10, $n_{Rps17,Gr64}$ = 10, 2-sided Mann–Whitney U test). (**I**) Wing discs heterozygous for *RpS23* (left panel) or both heterozygous for *RpS23* and homozygous null for *Gr64* (*Gr64af*) (right panel) assessed for cell death with a staining for cleaved-Dcp1 (red), along with quantification in (**J**) ($n_{Rps23}$ = 10, $n_{Rps23,Gr64}$ = 10, 2-sided Mann–Whitney U test). Horizontal lines indicate the mean in S1B, S1D, S1F and the median in S1H and S1J. Numerical data can be found in the "S1 Fig" sheet of S1 Data. Gr64, Gustatory Receptor 64; Rp, ribosome protein.
(TIF)

**S2 Fig. (Related to Fig 1). Characterisation of the Rps3 and Gr64 interaction in other imaginal discs.** (**A–C**) Representative images of tubulin flp-out clones (green) in an *RpS3* heterozygous background expressing a blank UAS control (**A**) or *Gr64f-RNAi* (**B**), assessed for cell death with a staining for cleaved-Dcp1 (magenta) along with quantification in (**C**) ($n_{UAS-control}$ = 10, $n_{Gr64f-RNAi}$ = 10, 2-sided Mann–Whitney U test). (**D–F**) *RpS3*$^{+/-}$ haltere discs expressing a blank UAS control (**D**) or *Gr64f-RNAi* (**E**) in the posterior compartment with *hh-Gal4*, stained with anti-Ci (cyan) to label the anterior compartment, and assessed for cell death with a staining for cleaved-Dcp1 (red) along with quantification in (**F**) ($n_{UAS-control}$ = 9, $n_{Gr64f-RNAi}$ = 10, 2-sided Wilcoxon signed rank test). (**G–I**) *RpS3*$^{+/-}$ leg discs expressing a blank UAS control (**G**) or *Gr64f-RNAi* (**H**) in the posterior compartment with *hh-Gal4*, stained with anti-Ci (cyan) to identify the anterior compartment, and assessed for cell death with a staining for cleaved-Dcp1 (red) along with quantification in (**I**) ($n_{UAS-control}$ = 13, $n_{Gr64f-RNAi}$ = 19, 2-sided Wilcoxon signed rank test). (**J, K**) Eye discs heterozygous for *RpS23* (**J**) or eye discs heterozygous for both *RpS23* and *Gr64* (**K**) assessed for cell death with a staining for cleaved-Dcp1 (red) along with quantification in (**L**) ($n_{Rps23}$ = 11, $n_{Rps23,Gr64}$ = 13, 2-sided Mann–Whitney U test). Horizontal lines indicate the median in all graphs. Numerical data can be found in the "S2 Fig" sheet of S1 Data. Gr64, Gustatory Receptor 64; RNAi, RNA interference; Rp, ribosome protein.
(TIF)

**S3 Fig. (Related to Fig 3). Effect of Gr64 inhibition in wild type and *RpS3*$^{+/-}$.** (**A, B**) GstD1-GFP reporter expression (green) in wild-type wing discs expressing *Gr64f-RNAi* driven by *enGal4* in the posterior compartment, stained with anti-Ci (grey) to label the anterior compartment, along with quantification in (**B**) ($n$ = 7, 2-sided *t* test). (**C–E**) Wing discs heterozygous mutant for *RpS3* without (**C**) or with (**D**) a heterozygous deficiency in the *Gr64* locus (ΔGr64) and assessed for GstD1-GFP reporter expression (green) along with quantification in (**E**) ($n_{RpS3}$ = 6, $n_{RpS3,ΔGr64}$ = 7, 2-sided Student *t* test, the horizontal line indicates the mean and

the whiskers reflect the 95% confidence interval). Numerical data can be found in the "S3 Fig" sheet of S1 Data. Gr64, Gustatory Receptor 64; RNAi, RNA interference; Rp, ribosome protein. (TIF)

**S4 Fig. (Related to Fig 3). Negative (no RNAi) controls.** (**A**) *RpS3*$^{+/-}$ wing discs carrying the *enGal4* driver but expressing no RNAi construct in the posterior compartment and stained for phosphorylated JNK (green) with quantification in (**B**) (*n* = 11, 2-sided paired *t* test). (**C**) *RpS3*$^{+/-}$ wing discs carrying the *enGal4* driver but expressing no RNAi in the posterior compartment and stained for phosphorylated eIF2α (green) along with quantification in (**D**) (*n* = 9, 2-sided paired *t* test). Numerical data can be found in the "S4 Fig" sheet of S1 Data. RNAi, RNA interference. (TIF)

**S1 Movie. (Related to Fig 4). Effect of Gr64 on calcium signalling.** Wing discs heterozygous for *RpS3*, expressing the calcium reporter *sqh-GCaMP3* ubiquitously and *Gr64f-RNAi* in the posterior compartment with *enGal4*. Gr64, Gustatory Receptor 64; RNAi, RNA interference. (AVI)

**S1 Data. Supporting information file.** Numerical data for all quantifications in the manuscript are organised per figure, where all quantifications relevant to a given figure are arranged in a separate sheet. For every experiment, repeat quantifications and statistical tests performed are included. (XLSX)

## Acknowledgments

We thank the Amrein lab and S.J. Moon for providing *Gr64 Drosophila* lines and T.E. Rusten for the ref(2)P antibody. We thank the Wolfson Bioimaging Facility at the University of Bristol, FlyBase [42], the Bloomington Drosophila Stock Center, and the Vienna Drosophila Research Center.

## Author Contributions

**Conceptualization:** Michael E. Baumgartner, Iwo Kucinski, Eugenia Piddini.

**Funding acquisition:** Eugenia Piddini.

**Investigation:** Michael E. Baumgartner, Alex Mastrogiannopoulos, Iwo Kucinski, Paul F. Langton.

**Methodology:** Michael E. Baumgartner, Alex Mastrogiannopoulos, Paul F. Langton.

**Supervision:** Eugenia Piddini.

**Writing – original draft:** Michael E. Baumgartner, Eugenia Piddini.

**Writing – review & editing:** Michael E. Baumgartner, Alex Mastrogiannopoulos, Eugenia Piddini.

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
