## [Editor Report · Decision Letter 0]

23 Sep 2021

Dear Dr Piddini, 

Thank you for submitting your manuscript entitled "Ribosome protein mutant cells rely on the GR64 cluster of gustatory receptors for survival and proteostasis in Drosophila" for consideration as a Short Report by PLOS Biology.

Your manuscript, reviews from eLife, and revision plan have now been evaluated by the PLOS Biology editorial staff as well as by an Academic Editor with relevant expertise and I am writing to let you know that we would like to consider a revised version of your manuscript that addresses the reviewer comments from eLife.

Before we can invite you to submit a revised manuscript, we need you to complete your submission by providing the metadata that is required for full assessment. To this end, please login to Editorial Manager where you will find the paper in the 'Submissions Needing Revisions' folder on your homepage. Please click 'Revise Submission' from the Action Links and complete all additional questions in the submission questionnaire.

We also think that your manuscript would be most appropriate for our “Discovery Report” format, and request that when completing your submission that you select the Discovery Report Article Type. As Discovery Reports have up to 4 figures, this would not require any formatting changes to your manuscript. For more information on our different article types, see here: https://journals.plos.org/plosbiology/s/journal-information#loc-discovery-reports

Once you have completed your submission, we will send you a formal "major revision" decision, which will include additional comments from the Academic Editor and a 3 month deadline for the revision.

Please re-submit your manuscript within two working days, i.e. by Sep 27 2021 11:59PM.

Kind regards,

Lucas Smith

Associate Editor

PLOS Biology

lsmith@plos.org

---

## [Editor Report · Decision Letter 1]

27 Sep 2021

Dear Dr Piddini,

Thank you for submitting your manuscript "Ribosome protein mutant cells rely on the GR64 cluster of gustatory receptors for survival and proteostasis in Drosophila" for consideration as a Discovery Report at PLOS Biology. As mentioned in our last email, your manuscript, the reviews from eLife, and your revision plan have been evaluated by the PLOS Biology editors and an Academic Editor with relevant expertise.

In light of the reviews from eLife, which I have appended below, we will not be able to accept the current version of the manuscript. However, we would welcome re-submission of a much-revised version that takes into account the reviewers' comments as outlined in your submission. Having discussed your revision plan with the Academic Editor, we think that it would be particularly important for your revised manuscript to provide data from the homozygous Gr mutants, plus the rescue experiments. Given that the core of the claims rely on genetic interactions, we think that such experiments would strengthen our confidence in the provocative claims reported here.

Additionally, after discussion within the team, we think that the framing of the study might be shifted a bit to be better suited for our Discovery Report format. If you agree, we would suggest the emphasis of the title and manuscript be shifted slightly to highlight the novel finding that GR64 cluster of gustatory receptors regulate proteotoxic stress and survival of non-neuronal cells, instead of presenting the findings as a modifier of the ribosome mutant protein stress phenotype. 

We cannot make any decision about publication until we have seen the revised manuscript and your response to the reviewers' comments. Your revised manuscript is also likely to be sent for further evaluation by the reviewers. We will do our best to engage with the same reviewers that initially reviewed the paper at eLife, however this is not always possible, as it depends on the reviewer availability. However, if we deviate from this commitment for external circumstances, we will let you know.

We expect to receive your revised manuscript within 3 months. 

**IMPORTANT - SUBMITTING YOUR REVISION**

*Re-submission Checklist*

*Published Peer Review*

*PLOS Data Policy*

*Blot and Gel Data Policy*

Sincerely,

Lucas Smith

Associate Editor

PLOS Biology

lsmith@plos.org

REVIEWS FROM eLife:

Reviewer #1 (Public Review):

This paper investigates mechanisms that modulate the phenotype of cells that are heterozygous for ribosomal protein genes (Rp/+). The previously described effect of the Rp/+ or minute genotype in completely heterozygous animals is delayed development and increased levels of proteostasis stress, which correlates with decreased autophagy and proteasome activity. In a clonal situation where Rp/+ cells are juxtaposed to wild type cells, they get eliminated by cell competition and apoptosis.

Based on results of transcriptomic analyses, the authors found that six clustered tasteGr64 receptor genes are more highly expressed in Rp/+ than in wild type cells. In loss of function experiments they show that heterozygous loss of the Gr64 cluster and even loss of heterozygosity for any one of these six genes or RNAi mediated knock down aggravates the stress response in Rp/+ cells and promotes apoptosis when these cells are in competition with neighboring wild type cells. Reporter experiments and immunostaining indicates that the Gr64, Rp double heterozygous cells experience higher levels of oxidative and unfolded protein stress. Heterozygosity or even complete loss Gr64 genes does not cause apoptosis of elevated stress markers in a wild type background.

This paper leaves a lot of questions open: What is the explanation of individual loss of function alleles for all six Gr64 receptors having the same dominant phenotype in an Rp/+ background? What is the molecular function of these receptors that modulates to the minute phenotype? Are any ligands involved? If so, which?

Reviewer #1 (Recommendations for the authors):

Figure 1 and supplemental Fig 1: Why do you show the experiment with the larger deficiency that requires complicated complementation with rescue constructs when you have a precise CRISPR deletion?

Figure 2a: why is no apoptosis detectable at the borders between WT and RpS3/+

clones? 

Some minor issues that should be addressed in the paper:

Line 108: "ΔGr64/ΔGr64 flies present with no known phenotypes other than a deficient gustatory responses" should be "than a deficient gustatory response" or "than deficient gustatory responses")

Line 129: "heterozygousity" should be "heterozygosity"

It would be helpful if the figures and supplemental figure were identified by the numbers that refer to them in the text.

Label the genotypes in the figures better, which drivers are used etc.

Fig 1E,F. It is not clear why the paper suggests that knockdown in Hh-expression domain which span most of a parasegment shows a cell-autonomous requirement for Gr64. In fact, there are Caspase positive cells outside the Hh-expression domain. Clonal knock down for example by MARCM and precise analysis at the clonal boundaries would be the accepted method to test cell autonomy.

Reviewer #2 (Public Review):

Baumgartner et al. follow up a curious observation they made previously (Kucinski et al. 2017) , in which they noticed that Drosophila larvae heterozygous for the ribosomal protein Rsp3 show upregulation of sugar taste receptor genes of the Gr64 subfamily (Gr64a-Gr64f). They test a functional involvement of these receptors by creating double

heterozygous Rsp3+/-, Gr64a-f+/-, larvae and analyzing imaginal discs for evidence of increased proteotoxic stress/apoptosis compared to Rsp3+/- imaginal discs and find the former have indeed an increase in expression of caspase3 and DCP1, stress/cell death markers. In a slightly modified experiment (Figure 2), they investigate whether this phenomenon is also seen in discs in which winner (Rsp3+/+) and loser cells Rsp3+/-) are placed adjacent to each other, finding that small groups of DCP1 expressing cells are indeed more frequent at the interface of winner/loser cells in heterozygous Gr64 discs, compared to Gr64+/+ discs. They develop a more nuanced assay, in which they downregulate Gr64s via RNAi, and by expressing this construct in the posterior part of the disc. In this assay, they observe a comparatively milder effect on apoptosis, assayed by several different markers. Employing this more sensitive assay, they explore whether Gr64a contributes to protein aggregation using two reporters that represent proteolytic pathways and poly-ubiquitinylated aggregates.

Reviewer #2 (Recommendations for the authors):

Strengths:

1) A non-canonical role for taste receptors in ribosome biogenesis is a very interesting finding. While no experiments to address the mechanism are presented, the observation presented here open up new lines of investigations.

2) The experiments are well-designed, and appropriate controls are included.

3) The paper is well-written and easy to digest. 

Weaknesses:

1) The authors should extend the analysis further, including phenotype assessment of the minute phenotype in hemi- and homozygous Gr64 mutant that are Rsp3+/-. Likewise, why did they never use homozygous Gr64 mutants in any of their experiments? The effects on the observed phenotype would be expected to be even stronger.

2) Interpretation of the data in Figure 2 is difficult. These mutations are mostly gene knock-ins into a locus that is transcribed from a polycistronic message, so any of these

mutations can affect levels of expression of the entire locus. While these knock-ins mostly the purpose to reveal cellular expression but using them as gene knock-outs is fraught with many potential caveats.

3) There are no "rescue" experiments in this paper. It appears that it would be very

valuable to see if the observed phenotypes (Capsase3, DCP1 expression) incurred by heterozygosity of the Gr64 locus can be rescued by expression of individual G64 genes.

4) The authors' previous data indicate that upregulation of Gr64 genes in Rsp3+/- (and

other heterozygous mutants of genes involved in ribosome biogenesis) larvae might be a counter measure by increasing intracellular Ca2+. They do not provide an opinion whether this would be a ligand dependent or independent process. A simple test for

that would be a rescue type of experiments with other Gr genes (see point 4) not related

to sugar receptors)

Minor points:

1) nomenclature for Gr64 mutants is not accurate and confusing (Gr64a-GAL4, Gr64b- LexA, etc). They should refer to the original nomenclature used by Fujii et al. (Gr64aGAL4 etc)

2) The authors initiate the investigation on a possible role of Gr64 genes based on differential expression (i.e. upregulation) of these genes in Rsp3+/- and other mutants. This was a very restricted analysis (wing imaginal disc). Is this phenomenon observed in other discs/tissues that form adult structures?

Reviewer #3 (Public Review):

I found this paper quite satisfying, because it is an excellent example of "following the science" to discover something new and totally unexpected about the way cells work. I thought the story was compelling, the presentation was clear and well written, and that the impact will be significant.

The authors followed up on observations made in a previous paper (Kucinski et al.,

2017), where they compared gene expression in wild type flies versus flies heterozygous for a mutation in a ribosomal protein gene or another gene, mahjong,

associated with proteotoxic stress. The analysis resulted in a long list of differentially expressed genes, some of which they explored in that paper. Here, they explore the reasons for the unusual upregulation of a family of gustatory receptors. Rather than representing a meaningless case of gene misregulation, they found that increased expression of these receptors is an adaptive response to proteotoxic stress. In this way, they implicated the receptors in regulating pathways that no one had suspected them of regulating.

I have no criticisms of the authors' choice of experiments, the methods they used, their interpretations or their presentation. The major strength of this paper is the soundness of the methods and results. It is an exceptionally "clean" and well-constructed report.

For me, the significance of the work relates to the long history of ribosomal protein gene haploinsufficiency in Drosophila and the increasing importance of Drosophila to understanding cellular responses to disrupted protein synthesis. A relatively short time ago, explanations for the Minute syndrome-the phenotypes seen in Drosophila when

one copy of a ribosomal protein gene is eliminated-were speculative. The syndrome was considered an esoteric "fly thing". Over the past decade or so, the phenotypes have been

tied to an assortment of cell stresses. A major challenge now is sorting through all the ways ribosome dysfunction affects cellular pathways. It is a complicated situation, but it mirrors cellular defects in human ribosomopathies and other diseases affecting protein synthesis. This paper is an excellent example of the relevance, importance and productivity of such studies: a new cellular pathway linking gustatory receptors to cellular stress responses has now been discovered from its contribution to the Minute syndrome.

Reviewer #3 (Recommendations for the authors):

After reading this paper, one is left with the impression that what you describe for reduce RpS3 dosage is universal to reduced dosage for all Rp genes. I would like to have seen some statement that interactions between RpS3 and the Gr64 genes might be RpS3 specific. Better yet, I would like to have seen at least some experimental evidence showing the same interactions with another Rp gene. While I understand why you think the effects are probably not RpS3 specific, I urge to be hesitant and modest in generalizing. I recommend including appropriate caveats in the text.

The text is very well written, but I had a hard time understanding the section on "flux" beginning with line 156. "Flux" is not a word that is universally understood. I had to go back to your previous paper to figure out what you were talking about and to understand the purposes of your ProteoFlux and ReFlux tools. For ease of reading, I strongly recommend that you use more familiar phrases such as "import anddegradation", "elimination", etc. and provide better descriptions of the ProteoFlux and

ReFlux tools.

Because an asterisk is used to denote an unknown allele, I wanted to know why you used RpS3[*] in the allele designation in the experiments associated with Figure 1, but I could find no description of the origin of the allele. From context, I suspect it came from Bloomington stock 5699. Please include a note about its origin somewhere (or, better yet, sequence it and the known alleles and identify it definitively).

On line 66, you meant to use "proximate", not "proximal".

You should use FlyBase nomenclature for the genetic elements in your experiments, especially transgenes, at least once so that readers will not have to guess the correspondences between your lab notations and the carefully curated FlyBase entries describing them. Shortened symbols are OK for the text, but standardized symbols should be included in the Materials and Methods. This should be done for the sake of accuracy and reproducibility and to show that you value the efforts of the people who work hard organizing and archiving genetic information for you. Along the same lines, you should acknowledge research resources such as FlyBase that have enabled your work by explicitly citing their grants.

---

## [Decision Letter · Decision Letter 2]

10 May 2022

Dear Dr Piddini,

Thank you for your patience while we considered your revised manuscript "The Gr64 cluster of gustatory receptors promotes survival and proteostasis of epithelial cells in Drosophila" for publication as a Discovery Report at PLOS Biology. This revised version of your manuscript has been evaluated by the PLOS Biology editors, the Academic Editor and two of the original reviewers. 

The reviews are appended below. As you will see, both reviewers think the findings reported here are interesting and Reviewer 3 is fully satisfied by the revision. However, Reviewer 1 has highlighted a few lingering concerns, including that the study does not fully elucidate the functional basis for the phenomena reported here. Reviewer 1 also highlights that some of the conclusions should be toned down.

Having discussed Reviewer 1’s concerns with the Academic Editor, we think that additional mechanistic studies would be interesting, but are beyond the scope of the current study, which we think fits the criteria for our Discovery Report format. Therefore, we would not require additional experimental data for publication at PLOS Biology. However, before we can accept your study we think it would be important for you to thoroughly address the other concerns raised by Reviewer 1 by toning down claims regarding cell autonomy in the abstract and manuscript and by adding further discussion of caveats, limitations, and future directions.

**IMPORTANT: Please also make sure to address the following data and other policy-related requests.

1 - Please provide a blurb which (if accepted) will be included in our weekly and monthly Electronic Table of Contents, sent out to readers of PLOS Biology, and may be used to promote your article in social media. The blurb should be about 30-40 words long and is subject to editorial changes. It should, without exaggeration, entice people to read your manuscript. It should not be redundant with the title and should not contain acronyms or abbreviations.

2 - DATA REQUEST: You may be aware of the PLOS Data Policy, which requires that all data be made available without restriction: http://journals.plos.org/plosbiology/s/data-availability. For more information, please also see this editorial: http://dx.doi.org/10.1371/journal.pbio.1001797

- Supplementary files (e.g., excel). Please ensure that all data files are uploaded as 'Supporting Information' and are invariably referred to (in the manuscript, figure legends, and the Description field when uploading your files) using the following format verbatim: S1 Data, S2 Data, etc. Multiple panels of a single or even several figures can be included as multiple sheets in one excel file that is saved using exactly the following convention: S1_Data.xlsx (using an underscore).

- Deposition in a publicly available repository. Please also provide the accession code or a reviewer link so that we may view your data before publication. 

Fig 1C,F,H; Fig 2H-I; Fig 3B,D,F,H; Fig 4C,IJ

Fig S1B,D,F,H,J; Fig S2C,F,I,L; Fig S3 B,E; Fig S4B,D

**Please also ensure that figure legends in your manuscript include information on where the underlying data can be found, and ensure your supplemental data file/s has a legend.

**Please ensure that your Data Statement in the submission system accurately describes where your data can be found

3) Please note that per journal policy, the model system/species studied should be clearly stated in the abstract of your manuscript. 

We expect to receive your revised manuscript within two weeks. 

*Published Peer Review History*

*Press*

Sincerely,

Lucas

Lucas Smith, Ph.D.,

Associate Editor,

lsmith@plos.org,

PLOS Biology

REVIEWER COMMENTS

Reviewer #1: Dirk Bohmann (note this reviewer has signed his review)

Review of PBIOLOGY-D-21-02398R2

Baumgartner et al.

This paper describes a previously unrecognized phenotype of Drosophila mutants that are deficient for a certain group of taste receptors. Six Gr64 receptors are expressed from a polycistronic mRNA. Flies that are heterozygous for mutations in one of more of these 6 clustered genes display increased levels of oxidative stress and cell death in tissues that are also minute, i.e. heterozygous for ribosomal protein (rp) gene loss of function mutations. This combined rp+/-, Gr64+/- phenotype is observed regardless of whether the double heterozygous genotype is shared by all cells in the organism or whether the mutants are in a competitive situation next to wild type cells, in which case cell competition is expected.

All three reviewers of the first submission of this manuscript agree that the phenomenon described in is interesting and unexpected. It may give rise to new lines of research that may help to better understand the intriguing biology of minute mutants. 

The major concern that reviewer 2 and I pointed out was that the paper describes in interesting observation but does not go very far in providing insight into the functional basis for the unexpected interplay between smell receptors and minute mutants. In the revised version one experiment is added which indicates that in rp/+ imaginal discs cells in which Gr64 expression is knocked down calcium signaling is reduced. Demonstration of a connection between this effect and the observed phenotype cell stress phenotype would require addional experiments. At this stage such a connection is a matter of speculation. So, the question of novelty and general interest remains.

The specific issues that I raised in my previous review included: 

It is difficult to interpret the data presented in figure 2 where individual insertion mutants in the each of the six Gr64 receptors are tested for their effect on rp/+ dependent cell death. Reviewer 2 expressed a similar concern. The authors point out, correctly, that it is difficult to know whether mutations in the coding regions of one of the six genes would only affect the expression that particular gene product or also that of some or all of its neighbors. In the end I am still confused about how to interpret this result, beyond: these are six more independent alleles that can be presumed to affect Gr64 function in a somewhat undefined way that support the conclusion of the previous picture.

I also wondered how clear it is that the effect of the rp+/-, Gr64+/- genotype is cell autonomous. There is clearly more cell death in the tissue regions in which Gr64 expression is reduced. But there is also some in adjacent areas. The newly added experiments looking at haltere and leg imaginal discs do not clarify this question. The conclusion of cell autonomy should be qualified.

The other issues I pointed out were mostly suggestions to improve the paper.

Reviewer 2 has some additional technical comments, several of which were satisfactorily addressed:

* Test minute mutations other than Rps3

This is satisfactorily addressed by showing experiments on RpS17 and Rps23 (revised Fig S1)

* Test the interaction of minutes with a Gr64 homozygous loss of function genotype. 

This was done, in an experiment with Rps23 (revised Fig S1). The presented results are consistent with the author's model.

* Do a rescue experiment, where expression Gr64 receptors would revert the mutant phenotype.

This was done using a UAS construct that has enough expression without a Gal4 driver. The presented results support the authors conclusion and are responsive to reviewer 2's query.

In conclusion

The revision provides additional lines of experimental evidence in support of the proposed role of Gr64 receptors in the minute phenotype and cell stress responses. It does however not offer more conclusive functional insight into the mechanism underlying this phenomenon.

Reviewer #3: I was asked to review this revised manuscript (The Gr64 cluster of gustatory receptors promotes survival and proteostasis of epithelial cells in Drosophila) for PLoS Biology after reviewing the original version for eLife. I was Reviewer 3 in the eLife reviews and wrote generally positive comments with only minor suggestions for revision. I was impressed by the original manuscript and see that the revisions in this version have only strengthened the presentation. In my opinion, you have adequately addressed the concerns presented by the eLife reviewers. You have certainly addressed the relatively minor concerns that I had. 

To me, this manuscript fulfills all the requirements for publication as a Discovery Article. It presents a significant discovery that will be interesting to a wide range of scientists. It will be particularly interesting to researchers interested in ribosomopathies and cellular stress. The results are novel and they are presented and analyzed with appropriate rigor and clarity. The manuscript is well written and pleasant to read. I see no need for additional experiments or significant revisions.

I am personally intrigued by this line of research. I think the connection you have made between ribosome dysfunction and gustatory receptor signaling is quite interesting. I am impressed that you followed up on your preliminary observations regarding the Gr64 genes in Minute mutants and found a new cellular role for this receptor gene family.

---

## [Editor Report · Decision Letter 3]

2 Jun 2022

Dear Dr Piddini,

Thank you again for your submitting a revised version of your manuscript "The Gr64 cluster of gustatory receptors promotes survival and proteostasis of epithelial cells in Drosophila" for publication as a Discovery Report at PLOS Biology. Your revision has been evaluated by the PLOS Biology editors and the Academic Editor, and we are largely satisfied by the changes made in response to the reviewers and to our previous editorial requests. However, in looking through the most recent revision, the Academic Editor has noticed two issues which we think should be addressed at this stage in order to make sure the manuscript is as solid as possible.

Therefore, before we can accept your study, we request that you address the following two comments from the Academic Editor in another revised manuscript.

COMMENTS FROM THE ACADEMIC EDITOR

1) Regarding my first comment on the previous version about the values in Figure 1A: I was mistaken in stating the "values are actually "log FC"". I meant to write in the Kucinski 2017 study Supp Table 1, the values are "Fold change" (i.e. neither log10 nor log2). The authors should double check this and correct.

2) The phenomenon of calcium flashes (Figure 4J) is intriguing as a potential physiological link between Gr64s and downstream cellular events, but as such flashes have not been described in the literature before - as far as I can see - it would be good to provide some context as to what the authors think they represent (from the video they seem to occur somewhat randomly in different cells/groups of cells in the disc). In addition, could the authors confirm that the small, but significant A-P asymmetry in frequency observed is really due to the Gr64 RNAi in the P compartment and not reflecting an inherent difference in calcium flash frequency in these two different regions of the disc (i.e. without any Gr64 RNAi)? If they do not have such data, they should at least mention this caveat.

We expect to receive your revised manuscript within two weeks. 

- a Response to Reviewers file that provides a detailed response to the reviewers' comments (if applicable) (if your cover letter describes the changes made to the requests above, you can upload a blank document here)

*Published Peer Review History*

*Press*

Sincerely,

Luke

Lucas Smith, Ph.D.

Associate Editor,

lsmith@plos.org,

PLOS Biology

---

## [Editor Report · Decision Letter 4]

14 Jun 2022

Dear Dr Piddini,

Thank you for the submission of your revised Discovery Report "The Gr64 cluster of gustatory receptors promotes survival and proteostasis of epithelial cells in Drosophila" for publication in PLOS Biology. On behalf of my colleagues and the Academic Editor, Richard Benton, I am pleased to say that we can in principle accept your manuscript for publication, provided you address any remaining formatting and reporting issues. These will be detailed in an email you should receive within 2-3 business days from our colleagues in the journal operations team; no action is required from you until then. Please note that we will not be able to formally accept your manuscript and schedule it for publication until you have completed any requested changes.

PRESS

Sincerely, 

Lucas Smith, Ph.D.

Associate Editor

PLOS Biology

lsmith@plos.org